# COVID-19 Pandemic Affects Children’s Education but Opens up a New Learning System in a Romanian Rural Area

**DOI:** 10.3390/children10010092

**Published:** 2023-01-02

**Authors:** Oana Miruna Oprea, Iuliana Elena Bujor, Alexandra Elena Cristofor, Alexandra Ursache, Bianca Sandu, Ludmila Lozneanu, Cristina Elena Mandici, Andreea Silvana Szalontay, Marcel Alexandru Gaina, Daniela-Roxana Matasariu

**Affiliations:** 1Department of Educational Science, “Ion Creanga” State Pedagogical University, MD-2069 Chisinau, Moldova; 2Department of Obstetrics and Gynecology, University of Medicine and Pharmacy ‘Gr. T. Popa’, 700115 Iasi, Romania; 3Department of Obstetrics and Gynecology, Cuza Vodă Hospital, 700038 Iasi, Romania; 4Department of Morpho-Functional Sciences I—Histology, Pathology, “Grigore T. Popa” University of Medicine and Pharmacy, 700115 Iasi, Romania; 5Psychiatry, Department of Medicine III, “Grigore T. Popa” University of Medicine and Pharmacy, 700115 Iasi, Romania; 6Institute of Psychiatry “Socola”, 36 Bucium Street, 700282 Iasi, Romania

**Keywords:** children, adolescent, COVID-19 pandemic, education, health, teachers, e-learning

## Abstract

(1) Background: In the context of the COVID-19 pandemic, the educational system in Romania faced major challenges. Online knowledge development was necessary and mandatory during this time; (2) Methods: our study included a group of 140 preadolescents and used a phenomenology qualitative method to investigate if the novel online teaching, implemented in a hurry during the pandemic without any previous teacher training, managed to replace face-to-face teaching; (3) Results: The students have expressed their joy for online courses, as long as they are kept interactive. Even though they feel nervous and worried when it comes to evaluation, the students claim they feel capable to learn all of the learning materials. Most of them are pleased by classes and do not get bored during them, feeling constantly motivated to actively participate in dialogue; (4) Discussions: Despite the lack of teaching-method standardization, our learning providers succeeded in accomplishing their tasks during online courses. Even in remote rural areas, they managed to assure the means for these children to access and take part in online courses; (5) Conclusions: our learning system must offer teachers the possibility to emphasize online education using adequate training programs aiming to develop technical and online pedagogical skills.

## 1. Introduction

SARS-CoV-2 infection was reported for the first time in Wuhan, Hubei Province in China. It represents an acute and severe respiratory infection that has rapidly spread worldwide, leading shortly to a pandemic, thus forcing governments, companies and educational institutions to take quick and drastic actions to reassure the safety and health of the population [1].

In the last 100 years, there has been no such challenge as the one provoked by the new SARS-CoV-2 coronavirus, with the whole world facing unprecedented situations regarding economics and society in terms of healthcare, with physical distancing as a premise for the limitation of the virus’s spread [2].

The school network in Romania is mostly the same as it was before 1990, and it is characterized by overcrowded classrooms, mainly in the urban areas, where in a normal classroom, from 30 up to 40 students can be found. Moreover, functional illiteracy places Romania on the last ranks in the European Union, as 40% of the 15-year-old students do not know how to use the information studied at school as an attribute for solving day-by-day problems [3]. Moreover, the forecasts show an additional 10% at the present generation of students, compared with the precedent one, as a result of closing the schools and moving the didactic process online [4,5].

### 1.1. COVID-19 Pandemic in Our Country

On March 16, a state of emergency was established by presidential decree, which considerably limited the mobility of the population, imposed restrictions on land and air transport, imposed quarantine measures for Romanian citizens coming from abroad, and extended the period in which the schools were closed. In fact, they continued to remain closed even after the end of the emergency state (13 May 2020) until the end of the school year, when the teaching staff and students were obligated by the order issued by the Ministry of Education to carry out the process of education in an online environment, from April [6,7,8].The three months of online education at the end of the previous school year (2020) were a good opportunity to identify deficiencies in the education system, as a premise for correcting them during the summer vacation in order to adapt activities to the challenges of the “school in a pandemic”. Mainly, they were based on three major principles:
Ensuring the technical infrastructure and qualified personnel for the eventuality of teaching in the online system;Ensuring the necessary conditions for the physical distancing between students in classrooms;Ensuring the necessary support so that parents could supervise minors during school activities at home [6,8].

### 1.2. Teaching Students

The teacher is the central character in the classroom. He is the student’s source of information, a model for learning social skills, and building various types of relationships. The teacher’s presence eases communication between students, shapes and manages the courses, while also crystalizing the essential information, tracing the way to use all these acquisitions in practice. A teacher must be able to deliver instructions that come as a help to the student, to prepare and offer effective lessons, to collaborate with students, and offer them feedback, being able to cover the curriculum with effectiveness [9].

### 1.3. The Learning System in Our Country

Even if we succeeded in partially transforming our learning system and exceeding the communist era, there are still many things that persist from those times. We still have crowded classrooms, with too many students to ensure the proper development of courses, a proper way of helping them acquire information and ensure an appropriate evaluation of the notions that they gained during class. The curriculum remains too large, with many unnecessary notions that a student is supposed to acquire. Sometimes, the explanations are hard to grasp for the student because the difficulty surpasses their ability to understand it. We are in the course of implementing a student-centered type of learning in trying to ensure a better understanding of the notions. Online courses during the COVID-19 pandemic only made it harder for both teachers and students, by requesting maximum adaptability to new teaching and learning methods in a very short period of time. These aspects motivated us to conduct our study [3,4,5].

### 1.4. Online Courses

Online knowledge development was necessary and mandatory during the COVID-19 pandemic, when the whole world was affected, with implications in all aspects of life, education being especially affected by it. Online education is defined as learning that takes place partially or totally via the Internet [10,11], indicating that the results of the students that followed online courses were influenced by the design features of the courses.

Previous research [10,12] tells us that an online course is defined as one whose content is 80% taught online. Face-to-face interaction includes courses that have 0–29% of the content taught in a virtual matter. This last category includes both classic courses and digital courses. A third way of teaching, called mixed or hybrid, has between 30% and 80% of the course content taught in an online matter. Generally, online courses are shared via the Internet [13]. Online courses can use digitalization to involve students by using animations, simulations, video clips, audio files, documents, and other interactive contents [14,15].

Online courses can be a good approach in offering students an insight into captivating topics, and some can solicit students so that they develop a certain grade of skillfulness as they go through them [15].

Online education has become popular because of its potential to offer more flexible access to various contents and instructions, by raising the learning possibilities for those who cannot or would not choose a traditional school, by getting together and disseminating the contents in a more facile way [16].

Online learning mirrors the way a teacher teaches, and it can be in synchrony (a type of communication in which the participants interact in the same time and space as a video conference) or in asynchrony (a type of communication that is divided in matter of time, like emails or online discussion forums) and accessed from various locations (inside or outside the school building).

By digitalization, the spectrum of courses that become available for students can be broadened, especially for those that come from rural areas or small towns, improving also the computer skills of the learners [17].

The Social learning Theory states the fact that new types of behavior result from student–teacher interaction, student–student interaction, or by observing teachers’ behavior. The actual impact of online teaching and learning due to the COVID-19 pandemic remains unquantified [18].

Our study differs from other studies, due to focusing on secondary school students, not very familiarized with online teaching, because this education sector benefits from scarce research regarding this aspect.

## 2. Participants and Methods

Our study included a group of 140 preadolescents registered in the secondary school cycle in Oniceni Secondary School from Suceava.

This study used a phenomenology qualitative method, as an attempt to find the way the students understand the experiences they have lived regarding the reshaping of education to an online type of learning.

The students enrolled in the study by voluntary choice, after they had the written consent of a parent/legal guardian. They were anonymous during the whole study, and they were assured that their answers’ confidentiality will be maintained. They were encouraged to give the most accurate answers and to express their thoughts and feeling regarding the questions.

The questionnaire was handed to the students in the year after the COVID-19 pandemic, by their schoolteachers, during coordination classes (Table 1). At the time when the questionnaire was handed out, the school was out of lockdown due to SARS-CoV-2 virus for 3 months, with the students being back on classes for one month.

The questionnaire was approved to be given for evaluation by the Ethical Committee of Oniceni Secondary School from Suceava (no. 1428 from 29 September 2021). We used a modified metacognition questionnaire after translating it into the Romanian language and validating it. The questions were translated into the Romanian language by three translators, and we have selected the best version for each item.

Our questionnaire includes 98 questions that are divided in 3 parts, in an attempt to underline the impact of the SARS-CoV-2 pandemic on the educational process. The first part includes general questions that refer to the social status, home environment and school results for the last semester of the student. The second and third parts are synthesizing students’ impressions over online courses and their feelings regarding interaction with other students and teachers. To be valid, all the 98 questions needed to be answered. All the questions from part two of our questionnaire were evaluated from 1 to 7, with1 meaning total disagreement and 7 complete agreement with the statement. In part three of our questionnaire, all the statements were evaluated from 1 to 5, with 1 meaning total disagreement and 7 complete agreement with the statement.

Data Analysis

We analyzed our data using both qualitative and quantitative methods. The descriptive statistics were analyzed and expressed as mean and frequencies (%). We evaluated all the answers obtained, performed frequency analyses, and provided the students’ exact words to increase the clarity and accuracy of their perception of the pandemic. Our approach to the qualitative content analysis was inductive.

We used Microsoft Excel Cronbach’s Alpha formula to evaluate the reliability of our questionnaire, and then we calculated the triangulation for the more important questions answered by our students from part II and III. When we estimated the saturation for our population of secondary school students, we observed that there were no new data obtained with the help of our questionnaire after 137 students responded (with a confidence level of 95%). We have gathered 140 complete questionnaires answered from a total population of 145 secondary school students.

## 3. Results

### 3.1. Part I

The questions we have included in the first part of our evaluation included information about the students’ socio-economic environment. Our preadolescents were both boys (54%) and girls (46%), that resided in the rural area, and that came both from low-, medium-, and high-income families. Eight of them were orphans of one/both parents or came from orphanages. From a total of 140 students, only 7 (5%) have failed to finish the last semester, including 6 in one subject and 1 in 3 subjects.

### 3.2. Part II

The reliability of the second part of our questionnaire was 0.887, meaning that there is a very good reliability between the various items.

The students have expressed their joy for online courses, as long as they are kept interactive (Table 2).

Most of the students were satisfied by the teaching materials, considering them interesting. Most preadolescents consider that, first of all, it is important to understand the lessons they are taught, even though in a given moment, an idea from the course does not make sense. For this purpose, when it applies, they search information from additional sources that are easier to understand.

The participants do not expect to have maximum results in the final exams, but are willing to learn from their mistakes and make progress. One third of the students are firmly convinced that the assimilated information will lead them to good and very good outcomes in the final exams and practical use in the future.

In the learning process, when studying for a test, most preadolescents try to corroborate the information from classes and books, by constantly applying self-assessment, even using their own words, to be sure they withheld the information, calling on old knowledge to learn new ones. When these are too difficult, most students choose either to quit studying or to just limit themselves on learning the essentials. The same thing does not happen when the materials presented in class are very dense and difficult, in which case the students try to learn it in its integrity.

Most preadolescents consider they own better learning skills compared to their classmates, but they do not consider being better than them or having better results.

Even though they feel nervous and worried when it comes to evaluation, the students claim they feel confident with themselves, and they visualize themselves being capable to learn all of the learning materials provided during online courses, even the ones they are not fond of. The students are organized; they develop study plans and rehearse the material multiple times before testing.

### 3.3. Part III

The reliability of the second part of our questionnaire was 0.893, meaning that there is a very good reliability between the various items.

The students seem to be relaxed and not anxious regarding the online courses. Most of them are pleased by classes and do not get bored during them, feeling constantly motivated to actively participate in dialogue (Table 3).

When they understand the online course and they intervene during it, the students feel confident and proud of themselves. They do not avoid asking questions about the course (Table 3).

The preadolescents do not consider that time used for studying as lost. Most of them think that online evaluation is a positive experience. Just like the ones face-to-face, online examination causes anxiety for students before it takes place, and after it finishes, the students experience feelings of ease and relief. Most students claimed they are optimistic regarding the results of online evaluation (Table 3).

The teacher’s evaluation method was appreciated and considered correct.

As a result, we can easily observe that their averages during the pandemic were slightly better compared to the ones during face-to-face courses (Table 4 and Table 5).

We have evaluated triangulation for the most relevant questions from part II and III of our questionnaire (Table 6).

## 4. Discussion

The COVID-19 pandemic has crystalized the need to reshape teaching methods in the context of social distancing to prevent the spread of the virus. Online teaching became the basis for continuing the learning process, with teachers finding themselves in the posture of adapting their methods for this new type of education. Though college teachers were partially familiarized with this type of teaching, it represented a novelty for primary and secondary school teachers in our country.

This challenge was sometimes perceived to have a negative impact on teachers’ pedagogical experience and life in Watermeyer et al.’s [19] study from universities in the United Kingdom, but was described as a success in Loima’s Finland and Basilaia et al.’s studies [20,21].

The situation was quite different in low-income and middle-income countries such as Romania. This aspect was mostly due to a lack of appropriate infrastructure, and lack of resources, especially financial resources, to support this type of education. Many children from rural areas did not have the means to participate in online classes (devices or internet access). These aspects were underlined also by Putra et al.’s 2020 Indonesian study [22]. There were many technical difficulties that hindered access to online education in our country.

Despite being from a middle-income country, with many technical difficulties to overcome, Romanian students tried to overcome and adapt to the newly solicitant learning situation. The results were even better than those obtained after face-to-face teaching at the end of 2019, immediately after the beginning of the pandemic.

Other aspects to consider are the disciplinary differences and the possibility to yield to online teaching. Primary and secondary school students are in the process of learning how to learn on their own. This is a possible explanation to why there are visible differences between how a college student perceived the online teaching experience and how primary and secondary school students considered it. Many studies, such as Agarwal et al., describe the fact that college students found this shift in teaching very useful, time-saving, and found it easy to obtain teaching materials [23]. The fact that it was a hurried-up procedure, due to lack of other options, without any previous pedagogical and technical training, contributed to inherent difficulties encountered by both teachers and students, especially from junior highs as detected in Fauzi et al.’s 2020 survey that evaluated the situation in 45 elementary schools [24]. Both teachers and students had to leave their comfort areas in a collective effort to continue their studies during the pandemic. The lack of guided study with the need for self-education effort or sustained by unprepared parents negatively affected school performance [25].

There were no standards established to guide online courses due to the fact that it was not a standardized universally accepted method of teaching, and the urgent need was to implement it fast during the pandemic. Thus, the method did not benefit from a detailed evaluation from both students and teachers before implementation [26].

Despite the lack of teaching-method standardization, our learning providers succeeded in accomplishing their tasks during online courses. Even in remote rural areas, they managed to assure the means for these children to access and take part in online courses. We need to appreciate both teachers’ and students’ efforts to quickly adapt to the new situation and to succeed in obtaining such results.

Even if one might consider online teaching as difficult, challenging, hollowed by human interaction and social formation, this method offers plenty of advantages. It provides the possibility of easy learning-material access, interactive activities, the shift from a hearing-focused learning to a visual one, and also, most importantly, is time-saving. But to obtain qualitative online education we need to learn from these two years of pandemic’s major challenges and improve this novel way of teaching by training our education providers in suitable pedagogic approaches and technological options and possibilities [9].

The way teachers reflect and perceive online courses mirrors in students’ performance and motivation. Their interest is maintained by the teachers’ ability to properly conduct such a challenging course, with the lack of direct face-to-face interaction being the most reported obstacle by education providers. Moreover, teachers do not have the possibility to accurately evaluate students during the learning process, in trying to monitor their progress [27,28].

Besides being time-saving, online education has the advantage of offering study possibilities for students from remote and isolated places. This aspect might have a major impact in lowering illiteracy in many countries, thus representing a strongly viable alternative [29].

Even if at the beginning of the pandemic, the teachers were not prepared to adopt online teaching techniques due to a lack of digital skills, studies reveal that the method succeeded in providing education in the pandemic social-isolation context [28,30].

## 5. Conclusions

It remains to be seen whether we will accept this method of teaching as being the main one in some fields of study, or use it as a complementary method of teaching, but only after we establish guidelines, prepare teachers accordingly, and solve/anticipate the related technical issues. The need to evaluate this type of teaching experience arose, due to the fact that it offered the best and only solution to continue the learning process in a social-distancing situation. We need to be prepared and anticipate future challenges, with adaptation being one of the skills required in our evolution process. We must offer our new generations valid opportunities and alternatives in a continuously changing environment. Our learning system must offer teachers the possibility to emphasize online education using adequate training programs aiming to develop technical and online pedagogical skills.

## Figures and Tables

**Table 1 children-10-00092-t001:** The questionnaire applied.

Questionnaire		
Part I	Part II	Part III
Which grade are you (V–VIII)?	I prefer interactive courses that allow me to learn new things.	I like to attend classes.
Your sex is masculine or feminine?	Compared to my colleagues, I expect to do better in class.	I want to learn as much as possible during class.
You come from a rural or an urban area?	I feel so nervous during tests, that I cannot remember the concepts I learned.	I am motivated to attend classes because I find them interesting.
Do you come from a low-income family?	It is important for me to learn what we are taught in class.	I feel confident after attending the lessons.
Are you an orphan (one or both parents)?	I like what we are taught in class.	I feel confident because I understood the study material.
What are the results you have obtained at the end of the last semester?	I am sure that I can understand the concepts taught in the lesson.	The confidence that comes from the fact that I understood the course material motivates me.
Have you ever failed a subject?	I think that I will be able to use the knowledge acquired during the classes in other subjects as well.	I am proud of myself.
If you have failed, how many subjects have you failed?	I expect to do very well in the end-of-semester assessment.	I think I can be proud of myself considering my level of preparation in lessons.
	Compared to my peers, I consider myself a better student.	I am motivated by the good academic results I obtained at the seminar.
	I often choose to read from sources that are more difficult to navigate, even if it will take more time and effort.	I am angry when I have to study.
	I am sure that I can solve all the requirements of the exams with very good results.	My anger rises when I think about the time wasted on lessons.
	I am very nervous when I am evaluated.	I wish I had not attended the classes because it annoys me.
	I think I will get a good grade on the final assessment of the school year.	I feel anxious during classes.
	Even when I do not do very well in the exam, I try to learn from my mistakes.	Before starting a class, I ask myself if I will be able to understand the new information.
	I believe that what we are taught will help me in the future.	Anxiety causes me to miss classes.
	I think my learning skills are better than my peers.	I feel embarrassed when I ask questions in class.
	I think the information taught in class is interesting.	If I answer wrong, I feel really stupid and I would like to make myself invisible.
	Compared to my peers, I believe I have more in-depth knowledge of the subjects taught.	I frequently lose hope when I do not understand a lesson.
	I know that I am able to learn the material in class.	When I quit, I also lose the energy to attend class.
	I worry a lot about my knowledge assessments	Lessons bore me.
	It is important for me to understand a subject.	I often think about what I could have done while I was taking an uninteresting course.
	During a test I think about the mistakes I make.	I am impatient for the classes to end.
	When I study for a test, I try to put together information from courses and books.	I like to learn new information.
	When I work on a project, I try to remember what we were taught in class.	I am so happy about my progress that I feel motivated to continue studying.
	I self-assess to make sure I have learned the study material correctly.	When my school results are good, I feel energetic.
	I find it difficult to understand the main ideas in a course.	I feel confident when I understand the study material provided.
	When the study material is difficult, I either give up or just learn the essential parts.	I feel optimistic when I make progress with my studies.
	When I learn, I put important ideas into my own words.	I think I can be proud of my academic achievements.
	I always try to understand what the teacher is trying to convey even if it does not make sense in the moment.	Since I want to be proud of my achievements, I constantly motivate myself.
	When I am studying for a test, I try to remember as much of what I have learned before as possible.	Studying makes me feel nervous.
	I copy my class notes to help me remember the material better.	When I study for a long period of time, I get so nervous that I want to throw my study materials out the window.
	I try to study extra for lessons even if I am not asked to.	I get irritated and tense when I study.
	Even when the material in class is dull and uninteresting, I learn it all the way through.	I worry about whether I will be able to learn all the study material.
	I usually repeat the subject several times before the test.	I feel ashamed when I realize that I lack certain practical skills.
	Before I start learning, I make a study plan.	When someone discovers how little I know about a subject, I avoid making eye contact with that person.
	I use old knowledge to understand new ones.	I feel so resigned to the fact that I cannot understand the study material.
	I frequently find myself in a situation where I do not understand a lesson even though I have read it several times.	My lack of confidence exhausts me even before I start studying.
	I find it difficult to follow what is taught in lessons.	Studying for classes bores me.
	When I learn something new, I try to put together the acquired information.	Boredom during study causes me to daydream.
	While going through a course, I tend to stop from time to time and review what I have learned.	I prefer to postpone learning boring material until the next day.
	I repeat out loud what I have learned to help me memorize the material.	For me, taking a test is a pleasant experience.
	I highlight the chapters I need to learn.	Before taking a test, I feel impatient.
	I push myself to get a good grade even in subjects I do not like.	I am optimistic about my school results.
		Confidence in my knowledge motivates me to do my best on the test.
		After the test I feel liberated.
		After the test I feel a relief of breathing.
		I often feel annoyed by the teacher’s marking method.
		Sometimes I feel so anxious that I would rather not take the test.
		I feel embarrassed when I do not answer test questions correctly.
		I am starting to think that no matter how hard I try, I will not be able to pass the test.

**Table 2 children-10-00092-t002:** Part II: Students’ behavior in the online classes (%).

	1	2	3	4	5	6	7
1. Compared to my colleagues, I expect to do better in class.	8.57	7.14	10.71	17.85	28.57	16.42	10.71
2. I feel so nervous during tests, that I cannot remember the concepts I learned.	12.14	12.14	8.57	5.71	14.28	18.57	28.57
3. It is important for me to learn what we are taught in class.	3.57	7.85	7.85	9.28	7.14	22.85	22.85
4. I like what we are taught in class.	4.28	7.14	7.14	13.57	18.57	24.28	22.14
5. I am sure that I can understand the concepts taught in the lesson.	5.71	7.85	6.42	6.42	21.42	29.28	14.28
6. I think that I will be able to use the knowledge acquired during the classes in other subjects as well.	6.42	7.85	7.14	15.71	14.28	28.57	20
7. I expect to do very well in the end-of-semester assessment.	4.28	5.71	12.14	31.42	22.85	12.14	11.42
8. Compared to my peers, I consider myself a better student.	11.42	6.42	17.14	18.57	20	16.42	10
9. I often choose to read from sources that are more difficult to navigate, even if it will take more time and effort.	9.28	9.28	12.85	24.28	18.57	16.42	9.28
10. I am sure that I can solve all the requirements of the exams with very good results.	5.71	6.42	11.42	26.42	22.14	20.71	7.14
11. I am very nervous when I am evaluated.	6.42	6.42	11.42	12.85	22.14	17.14	23.57
12. I think I will get a good grade on the final assessment of the school year.	6.42	6.42	8.57	24.28	21.42	22.85	10
13. Even when I do not do very well in the exam, I try to learn from my mistakes.	5.71	6.42	7.85	7.14	20	14.28	38.57
14. I think my learning skills are better than my peers.	8.57	6.42	15	4120	29.28	10.71	9.28
15. I think the information taught in class is interesting.	5	8.57	10.71	16.42	17.14	18.57	23.57
16. Compared to my peers, I believe I have more in-depth knowledge of the subjects taught.	6.42	10	13.57	23.57	20.71	16.42	9.28
17. I know that I am able to learn the material in class.	5	5	5.71	18.57	20	17.14	28.57
18. I worry a lot about my knowledge assessments.	3.57	3.57	5	9.28	38.57	13.57	26.42
19. It is important for me to understand a subject.	3.57	5.71	6.42	7.14	25.71	17.14	34.28
20. During a test I think about the mistakes I make.	4.28	5	7.14	5.71	15.71	25.71	36.42
21. When I study for a test, I try to put together information from courses and books.	3.57	5	7.14	5.71	10	17.14	51.42
22. When I work on a project, I try to remember what we were taught in class.	4.28	6.42	7.85	5.71	7.85	20.71	47.14
23. I self-assess to make sure I have learned the study material correctly.	5	5	10	10	11.42	25.71	32.85
24. I find it difficult to understand the main ideas in a course.	17.14	15	7.14	10.71	22.14	16.42	11.42
25. When the study material is difficult, I either give up or just learn the essential parts.	11.42	8.57	13.57	16.42	19.28	17.85	12.85
26. When I learn, I put important ideas into my own words.	6.42	3.57	7.85	15	15	21.42	30.71
27. I always try to understand what the teacher is trying to convey even if it does not make sense in the moment.	3.57	3.57	8.57	14.28	18.57	27.14	24.28
28. When I am studying for a test, I try to remember as much of what I have learned before as possible.	7.85	6.42	7.14	12.85	10.71	16.42	38.57
29. I copy my class notes to help me remember the material better.	9.28	9.28	7.85	7.85	18.57	14.28	34.28
30. I try to study extra for lessons even if I am not asked to.	10.71	9.28	12.85	18.57	16.42	13.57	18.57
31. Even when the material in class is dull and uninteresting, I learn it all the way through.	10.71	16.42	7.14	17.14	17.85	22.14	15.71
32. I usually repeat the subject several times before the test.	3.57	7.85	10	10	17.14	17.14	34.28
33. Before I start learning, I make a study plan.	10.71	12.14	8.57	10	18.57	17.14	17.14
34. I use old knowledge to understand new ones.	9.28	12.14	10	17.14	14.28	14.28	22.85
35. I frequently find myself in a situation where I do not understand a lesson even though I have read it several times.	10.71	6.42	8.57	10	19.28	20.71	24.28
36. I find it difficult to follow what is taught in lessons.	9.28	17.85	17.85	23.57	17.14	10.71	10.71
37. When I learn something new, I try to put together the acquired information.	6.42	5	10	12.85	13.57	22.14	30
38. While going through a course, I tend to stop from time to time and review what I have learned.	7.85	12.14	7.14	14.28	12.14	23.57	24.28
39. I repeat out loud what I have learned to help me memorize the material.	9.28	5	9.28	11.42	17.85	15	34.28
40. I highlight the chapters I need to learn.	12.14	9.28	10	10	12.85	16.42	30.71
41. I push myself to get a good grade even in subjects I do not like.	5	5	10	7.14	15.71	25	30.71

**Table 3 children-10-00092-t003:** Part III: Students’ class involvement (%).

	1	2	3	4	5
1. I like to attend classes.	9.28	12.14	11.42	22.85	44.28
2. I want to learn as much as possible during class.	6.42	5.71	13.57	23.57	50.71
3. I am motivated to attend classes because I find them interesting.	7.14	7.85	9.28	11.42	21.42
4. I feel confident after attending the lessons.	8.57	8.57	14.28	20	48.57
5. I feel confident because I understood the study material.	6.42	6.42	14.28	29.28	43.57
6. The confidence that comes from the fact that I understood the course material motivates me.	7.85	7.14	17.14	19.28	41.42
7. I am proud of myself.	6.42	8.57	7.85	10.71	23.57
8. I think I can be proud of myself considering my level of preparation in lessons.	5.71	10	19.28	32.14	32.85
9. I am motivated by the good academic results I obtained at the seminar.	5.71	10	20	32.85	30
10. I am angry when I have to study.	27.14	21.42	17.14	14.28	17.14
11. My anger rises when I think about the time wasted on lessons.	30	18.57	12.85	24.28	14.28
12. I wish I had not attended the classes because it annoys me.	42.85	22.85	10	12.85	11.42
13. I feel anxious during classes.	25.71	22.85	21.42	17.14	12.85
14. Before starting a class, I ask myself if I will be able to understand the new information.	14.28	15.71	15.71	18.57	35.71
15. Anxiety causes me to miss classes.	48.57	12.85	11.42	14.28	12.85
16. I feel embarrassed when I ask questions in class.	30	20	15.71	18.57	14.28
17. If I answer wrong, I feel really stupid and I would like to make myself invisible.	17.14	18.57	15.71	17.14	31.42
18. I frequently lose hope when I do not understand a lesson.	20	21.42	17.14	18.57	21.42
19. When I quit, I also lose the energy to attend class.	24.28	22.85	17.14	18.57	15.71
20. Lessons bore me.	31.42	24.28	12.85	17.14	14.28
21. I often think about what I could have done while I was taking an uninteresting course.	23.57	23.57	20	14.28	18.57
22. I am impatient for the classes to end.	18.57	15.71	21.42	11.42	30
23. I like to learn new information.	12.85	11.42	12.85	22.85	40
24. I am so happy about my progress that I feel motivated to continue studying.	10	11.42	21.42	24.28	32.85
25. When my school results are good, I feel energetic.	10	11.42	12.85	14.28	51.42
26. I feel confident when I understand the study material provided.	10	10	17.14	22.85	40
27. I feel optimistic when I make progress with my studies.	11.42	14.28	20	17.14	37.14
28. I think I can be proud of my academic achievements.	5.71	7.14	11.42	34.28	41.42
29. Since I want to be proud of my achievements, I constantly motivate myself.	10	12.85	15.71	31.42	30
30. Studying makes me feel nervous.	31.42	22.85	17.14	15.71	12.85
31. When I study for a long period of time, I get so nervous that I want to throw my study materials out the window.	30	24.28	15.71	15.71	14.28
32. I get irritated and tense when I study.	30	21.42	15.71	18.57	14.28
33. I worry about whether I will be able to learn all the study material.	15.71	18.57	20	18.57	27.14
34. I feel ashamed when I realize that I lack certain practical skills.	16.42	15	22.14	20.71	29.28
35. When someone discovers how little I know about a subject, I avoid making eye contact with that person.	16.42	19.28	23.57	23.57	17.14
36. I feel so resigned to the fact that I cannot understand the study material.	22.14	22.14	20	19.28	16.42
37. My lack of confidence exhausts me even before I start studying.	29.28	19.28	20.71	20.71	13.57
38. Studying for classes bores me.	28.57	24.28	14.28	20	12.85
39. Boredom during study causes me to daydream.	33.57	18.57	17.85	18.57	11.42
40. I prefer to postpone learning boring material until the next day.	29.28	25	25	11.42	9.28
41. For me, taking a test is a pleasant experience.	27.14	9.28	14.28	30.71	19.28
42. Before taking a test, I feel impatient.	27.14	12.85	12.85	29.28	17.85
43. I am optimistic about my school results.	8.57	10	11.42	44.28	27.14
44. Confidence in my knowledge motivates me to do my best on the test.	7.85	7.14	17.85	35.71	31.42
45. After the test I feel liberated.	9.28	5.71	12.14	20	52.85
46. After the test I feel a relief of breathing.	6.42	6.42	9.28	24.28	53.57
47. I often feel annoyed by the teacher’s marking method.	32.85	17.85	18.57	14.28	16.42
48. Sometimes I feel so anxious that I would rather not take the test.	30	35.71	9.28	14.28	10.71
49. I feel embarrassed when I do not answer test questions correctly.	16.42	14.28	14.28	27.14	27.85
50. I am starting to think that no matter how hard I try, I will not be able to pass the test.	18.57	12.85	35.71	17.14	15.71

**Table 4 children-10-00092-t004:** Averages in second semester during COVID pandemic with online courses (%).

	10	9–9.99	8–8.99	7–7.99	6–6.99	5–5.99
5th grade	0	10.71	6.42	7.14	0.71	0
6th grade	0.71	6.42	10	8.57	2.14	3.57
7th grade	0	8.57	6.42	7.14	0.71	0
8th grade	0.71	7.14	2.85	6.42	1.42	2.14

**Table 5 children-10-00092-t005:** Averages in first semester without COVID pandemic, face-to-face courses (%).

	10	9–9.99	8–8.99	7–7.99	6–6.99	5–5.99
5th grade	0	8.57	6.42	7.85	1.42	0.71
6th grade	0	5	9.28	9.28	4.28	3.57
7th grade	0	7.14	5	7.85	2.14	0.71
8th grade	0	5	2.14	7.85	3.57	2.14

**Table 6 children-10-00092-t006:** Triangulation for results of the most relevant questions from part II and III of the questionnaire.

Part II: Students’ Behavior in the On-Line Classes	Triangulation	Part III: Students’ Class Involvement	Triangulation
I like what we are taught in class.	0.0018	I like to attend classes.	0.0026
I am sure that I can understand the concepts taught in the lesson.	0.0018	I want to learn as much as possible during class.	0.0026
I think that I will be able to use the knowledge acquired during the classes in other subjects as well.	0.0018	I am motivated to attend classes because I find them interesting.	0.0027
I expect to do very well in the end-of-semester assessment.	0.0018	I feel confident after attending the lessons.	0.0026
Compared to my peers, I consider myself a better student.	0.0018	I feel confident because I understood the study material.	0.0025
I am sure that I can solve all the requirements of the exams with very good results.	0.0018	I am motivated by the good academic results I obtained at the seminar.	0.0025
I think I will get a good grade on the final assessment of the school year.	0.0018	I feel anxious during classes.	0.0025
I think my learning skills are better than my peers.	0.0018	I feel embarrassed when I ask questions in class.	0.0025
I think the information taught in class is interesting.	0.0018	Lessons bore me.	0.0025
Compared to my peers, I believe I have more in-depth knowledge of the subjects taught.	0.0018	I like to learn new information.	0.0025
I know that I am able to learn the material in class.	0.0018	I am so happy about my progress that I feel motivated to continue studying.	0.0025
I worry a lot about my knowledge assessments.	0.0019	Since I want to be proud of my achievements, I constantly motivate myself.	0.0025
It is important for me to understand a subject.	0.0018	For me, taking a test is a pleasant experience.	0.0025
When I study for a test, I try to put together information from courses and books.	0.0019	Before taking a test, I feel impatient.	0.0025
I self-assess to make sure I have learned the study material correctly.	0.0018	I am optimistic about my school results.	0.0025
When the study material is difficult, I either give up or just learn the essential parts.	0.0018	Confidence in my knowledge motivates me to do my best on the test.	0.0025
I always try to understand what the teacher is trying to convey even if it does not make sense in the moment.	0.0018	After the test I feel liberated.	0.0026
When I am studying for a test, I try to remember as much of what I have learned before as possible.	0.0019	After the test I feel a relief of breathing.	0.0026
Even when the material in class is dull and uninteresting, I learn it all the way through.	0.0018	I often feel annoyed by the teacher’s marking method.	0.0025
Before I start learning, I make a study plan.	0.0018	Sometimes I feel so anxious that I would rather not take the test.	0.0025
I use old knowledge to understand new ones.	0.0018	I am starting to think that no matter how hard I try, I will not be able to pass the test.	0.0025
When I learn something new, I try to put together the acquired information.	0.0018		
I highlight the chapters I need to learn.	0.0019		
I push myself to get a good grade even in subjects I do not like.	0.0018		

## Data Availability

Not applicable.

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
