# Peer review of "COVID-19 Pandemic Affects Children’s Education but Opens up a New Learning System in a Romanian Rural Area"

_children, 2023, doi:10.3390/children10010092_

Round 1

Reviewer 1 Report

I found the conclusion to be very clear about whether the method of teaching is clear and concise. This study gives impactful information about how COVID-19 affected students and teaching. Great article and read.

I think the authors did a great job in the research and the presentation. No major changes as far as I can see.

Author Response

On behalf of all the authors, I would like to thank you for taking the time from your busy schedule to revise our manuscript.

Thank you for your evaluation.

Reviewer 2 Report

The research should be reviewed in terms of theoretical foundation. All considerations are included in the pdf.

Author Response

On behalf of all the authors, I would like to thank you for the constructive comments in trying to improve our work, and for taking the time from your busy schedule to revise our manuscript.

Please find the detailed point-by-point-response below.

Reviewer 3 Report

1- I wonder why it is needed to include the historical events of COVID19! It is enough to talk about education. 

2- Examples should be given on each domain of the questionnaire. 

3- Data analysis should be detailed. 

4- Not all the items in the behavior domain indicate the behaioural aspect. For example, the first item is about preference which is more affective than behavioural. The 15th is affective too. 

5- You say:  and has used a phenomenology qualitative method. The study seems quantitative and not qualitative. You should detail the methodology. 

Author Response

On behalf of all the authors, I would like to thank you for the constructive comments in trying to improve our work, and for taking the time from your busy schedule to revise our manuscript.

Please find the detailed point-by-point-response below.

  1. I wonder why it is needed to include the historical events of COVID19! It is enough to talk about education.

Response: As suggested we tried to reduce the information about COVID 19 pandemic and concentrated on education. We have left only some important aspects that crystalize the difficulties encountered by our education system during on-line teaching imposed by the social distancing during pandemic.

  1. Examples should be given on each domain of the questionnaire. 

Response: We have inserted the whole questionnaire in our article with examples from each of the three parts.

  1. Data analysis should be detailed. 

Response: We have detailed the data analysis.

  1. Not all the items in the behavior domain indicate the behavioral aspect. For example, the first item is about preference which is more affective than behavioral. The 15th is affective too. 

Response: As suggested, due to the fact that the two questions were considered to be more affective than behavioral, we have excluded them.

  1. You say:  and has used a phenomenology qualitative method. The study seems quantitative and not qualitative. You should detail the methodology. 

Response: We have detailed the methodology.

Thank you for your evaluation.

Round 2

Reviewer 3 Report

Thank you for the improvement. 

Still, the qualitative data analysis is not detailed. Trustworthiness, triangulation and suturation should be described. 

Author Response

We have detailed the statistical analysis and calculated the reliability of our questionnaire, triangulation and saturation.
